# Quantitative Structure–Electrochemistry Relationship (QSER) Studies on Metal–Amino–Porphyrins for the Rational Design of CO_2_ Reduction Catalysts

**DOI:** 10.3390/molecules28073105

**Published:** 2023-03-30

**Authors:** Furong Chen, Amphawan Wiriyarattanakul, Wanting Xie, Liyi Shi, Thanyada Rungrotmongkol, Rongrong Jia, Phornphimon Maitarad

**Affiliations:** 1Research Center of Nano Science and Technology, Department of Chemistry, College of Sciences, Shanghai University, Shanghai 200444, China; 2Program in Chemistry, Faculty of Science and Technology, Uttaradit Rajabhat University, Uttaradit 53000, Thailand; 3Emerging Industries Institute Shanghai University, Jiaxing 314006, China; 4Center of Excellence in Biocatalyst and Sustainable Biotechnology, Department of Biochemistry, Faculty of Science, Chulalongkorn University, Bangkok 10330, Thailand; 5Program in Bioinformatics and Computational Biology, Graduate School, Chulalongkorn University, Bangkok 10330, Thailand; 6Department of Physics, Materials Genome Institute, Shanghai University, Shanghai 200444, China

**Keywords:** metalloporphyrin, QSER, DFT, catalyst design, CO_2_RR

## Abstract

The quantitative structure–electrochemistry relationship (QSER) method was applied to a series of transition-metal-coordinated porphyrins to relate their structural properties to their electrochemical CO_2_ reduction activity. Since the reactions mainly occur within the core of the metalloporphyrin catalysts, the cluster model was used to calculate their structural and electronic properties using density functional theory with the M06L exchange–correlation functional. Three dependent variables were employed in this work: the Gibbs free energies of H*, C*OOH, and O*CHO. QSER, with the genetic algorithm combined with multiple linear regression (GA–MLR), was used to manipulate the mathematical models of all three Gibbs free energies. The obtained statistical values resulted in a good predictive ability (R^2^ value) greater than 0.945. Based on our QSER models, both the electronic properties (charges of the metal and porphyrin) and the structural properties (bond lengths between the metal center and the nitrogen atoms of the porphyrin) play a significant role in the three Gibbs free energies. This finding was further applied to estimate the CO_2_ reduction activities of the metal–monoamino–porphyrins, which will prove beneficial in further experimental developments.

## 1. Introduction

The combustion of fossil fuels has contributed to the accumulation of CO_2_ in the atmosphere, which has led to significant increases in the greenhouse effect [1]. To achieve a carbon-neutral economy, increasing efforts have been devoted to converting CO_2_ into fuel molecules or value-added chemical products [2,3]. Among numerous strategies, the electrochemical reduction of CO_2_ is particularly appealing due to its mild reaction conditions, its tunable external parameters, such as electrode potentials, and available operating devices [2,3,4]. In addition, it uses water as a hydrogen source, makes full use of environmentally friendly renewable energy sources (such as solar, tidal, and wind energy), and can simultaneously contribute to carbon recycling and utilization and renewable electricity storage [5,6]. However, challenges remain in the electrochemical CO_2_ reduction reaction (CO_2_RR) since CO_2_ is a linear molecule with strong C=O bonds and requires a large overpotential for its activation [5]. Moreover, current electrochemical catalysts consistently suffer from low Faradaic efficiency (FE), high overpotential, and poor current density because of the competing hydrogen evolution reaction (HER) [7,8,9]. Therefore, the rational design of electrochemical catalysts with excellent catalytic performance and selectivity is imperative.

Metalloporphyrin complexes have been used as catalysts for CO_2_ reduction on account of their unique structure and electronic properties [10,11,12,13,14,15,16]; moreover, by changing the ligands on porphyrin complexes, one can modulate the different catalytic activities of such complexes [17,18,19,20,21]. Liu et al. used density functional theory (DFT) to design a series of transition-metal–porphyrins (TM–PPs) and found that the TM–PP monolayers display excellent catalytic stability and electrochemical CO_2_ reduction selectivity [22]. In another study, Davethu et al. tested several DFT methods to elucidate the ligand effect in the CO_2_RR performance of iron–porphyrins, finding that second-coordination-sphere perturbations influence CO_2_ positioning on the metalloporphyrins [23]. Abdinejad et al. observed that Fe–amino–porphyrins can selectively reduce CO_2_ to carbon monoxide (CO) at ambient pressure and temperature with competitive turnover numbers (TONs) [24]. Further, Wu et al. investigated the electrochemical CO_2_ reduction performance of a heterogenized zinc–porphyrin complex and demonstrated that such an electrocatalyst delivers a turnover frequency as high as 14.4 site^−1^ s^−1^ and an FE as high as 95% for the electroreduction of CO_2_ to CO at −1.7 V versus the standard hydrogen electrode [25].

During the CO_2_ reduction reaction, the first protonation step is CO_2_ + H^+^ + e^−^ → C*OOH or O*CHO [26]. These two main intermediates (C*OOH and O*CHO) will lead to the products of CO and HCOOH, respectively [27,28]. In the meantime, a hydrogen evolution reaction (HER) will also occur and it will form H* intermediates; this reaction is a competitive reaction with CO_2_RR [29]. The three important intermediates are shown in Figure 1. In theoretical studies, the Gibbs free energy of H* formation, G(H*), has been used to determine the favorable reaction between the hydrogen evolution reaction and the CO_2_RR [30], while the Gibbs free energies G(C*OOH) and G(O*CHO) can be used to predict the favored products [31,32,33,34,35,36]. Thus, based on theoretical catalytic studies, G(H*), G(C*OOH), and G(O*CHO) can be calculated and compared to determine the product selectivity.

The quantitative structure–electrochemistry relationship (QSER) method can be used to study the relationship between the structures and electrocatalytic activity of catalysts and is widely applied in electrocatalysis research [37,38]. In other words, this approach can be used to correlate a catalyst’s activity with its structural and electronic properties. In this work, the concept of QSER with the genetic algorithm combined with multiple linear regression (GA–MLR) was applied to develop the mathematical models for the Gibbs free energies of the three intermediates (H*, C*OOH, and O*CHO), and DFT was used to investigate the structural and electronic properties of the metal–porphyrin catalysts. We first obtained a series of 10 TM–PPs (TM = Sc, Ti, V, Cr, Mn, Fe, Co, Ni, Cu, and Zn) and their G(H*), G(C*OOH), and G(O*CHO) values from the literature [22]. Subsequently, we used the collected data to train the QSER models and then used them to predict the product selectivity of the CO_2_RR for a series of newly designed transition-metal monoamino-substituted tetraphenylporphyrins (TM–Amino–TPPs, TM = Sc, Ti, V, Cr, Mn, Fe, Co, Ni, Cu, and Zn); the workflow is depicted in Figure 2.

## 2. Results and Discussion

### 2.1. Correlation between DFT Structural and Electronic Properties and Gibbs Free Energies

In this work, the structural and electronic properties were obtained, consisting of the natural bond orbital (NBO) charges of the TM (X1); average NBO charges of N_a_ and N_b_ (X2), C_a_ (X3), and C_b_ (X4); dipole moment (X5); polarizability (X6); average TM–N_a_ and TM–N_b_ bond length (X7); and average N_a_–N_b_ distance (X8), as shown in Figure 3 and Table 1. The correlation of the three Gibbs free energies and the eight DFT properties of the TM–PP complexes have also been analyzed using the Pearson correlation coefficient, as shown in Figure 4. It indicates that the average NBO charges of Ca (X3) have the highest positive correlation with G(H*) and G(C*OOH), with correlation coefficients of 0.731 and 0.812, respectively. On the other hand, the NBO charges of the TM (X1) and the average NBO charges of N_a_ and N_b_ (X2) show a higher negative correlation with G(H*) with correlation coefficients of −0.231 and −0.282, respectively. Further, the average NBO charges of C_a_ (X3) and C_b_ (X4) have a positive effect on the values of G(O*CHO), with correlation coefficients of 0.780 and 0.878, respectively; however, the NBO charges of the TM (X1), dipole moment (X5), polarizability (X6), average TM–Na and TM–N_b_ bond length (X7), and average N_a_–N_b_ distance (X8) have a negative effect on G(O*CHO), with correlation coefficients of −0.774, −0.503, −0.631, −0.576, and −0.626, respectively. This indicates that X1 and X3 can greatly affect the values of G(H*) and G(C*OOH), and while these two descriptors affect G(O*CHO), X4, X6, X7, and X8 also result in determining the values of G(O*CHO).

### 2.2. GA–MLR QSER-Model Relationship

The GA method was used to identify the molecular properties that were highly correlated with the three Gibbs free energies. The occurrences of the populations of the eight descriptors for the three models are depicted in Appendix A, indicating that G(H*) and G(C*OOH) are highly correlated with X1, X3, X7, and X8, while G(O*CHO) has a high correlation with X1, X2, X6, and X7. The obtained QSER equations to predict the Gibbs free energy values of H* (Y1), C*OOH (Y2), and (O*CHO) (Y3) for TM–PPs with MLR and GA–MLR methods are shown in Appendix A. Table 2 shows the MLR models of Y1, Y2, and Y3 are denoted as Equations (1A,2A,3A), respectively, while the selected GA–MLR models of Y1, Y2, and Y3 are represented in Equations (1B,2B,3B), respectively. The QSER models with the MLR method resulted in a low predictive ability of R^2^(CV), which cannot be used for the further prediction of new design catalysts. Whereas the GA–MLR technique could improve the statistical values of the predictive ability of R^2^(CV), these obtained QSER with GA–MLR models are used for predicting the new design of metal–monoamino–porphyrin catalysts.

QSER GA–MLR of G(H*) model. Equation (1B) (see Table 2) represents the QSER model obtained from the GA–MLR method for the prediction of G(H*). The model yields good statistical values, with R^2^ = 0.945, R^2^(CV) = 0.832, RMSE = 0.152, and the F-value = 34.273. Equation (1B), Y1 = G(H*) = 383.311 × ramp (X3 − 0.183) + 29.037 × ramp (X7 − 1.953) + 33.1439 × ramp (2.069 − X7) − 3.440, implies that X3 and X7 play an important role in the prediction of G(H*). Thus, the longer the average TM–N_a_ and TM–N_b_ bond length (X7) and the more positive the charges are on the TM (X3), the higher the G(H*) value will be. Considering Table 1, it is found that Mn has the smallest positive charge for C_a_, while Sc has the most positive charge of C_a_ and the longest TM–N bond length. This indicates that Sc may lead to higher G(H*) values, while Ni gives rise to lower ones.

QSER GA–MLR of G(C*OOH) model. The QSER model obtained from the GA–MLR method for the prediction of G(C*OOH) is given in Equation (2B) (Table 2), with R^2^ = 0.957, R^2^(CV) = 0.924, RMSE = 0.120, and the F-value = 44.384. Considering Equation (2B), Y2 = G(C*OOH) =11.025× ramp (X1 − 1.011) + 367.618 × ramp (X3 − 0.184) + 11.838 × ramp (1.768 − X1) − 8.726, X3 and X1 are mainly related to the G(C*OOH) values; the more positive the charges are on C_a_ (X3) and TM (X1), the higher the G(C*OOH). Considering the relative charges for the TM and C_a_ for the various TM species, as shown in Table 1, Sc leads to higher G(C*OOH) values, while Mn gives rise to lower ones.

QSER GA–MLR of G(O*CHO) model. For the prediction of G(O*CHO), the QSER GA–MLR model is given by Equation (3B) (see Table 2). The statistical results of R^2^, R^2^(CV), RMSE, and the F-value are 0.955, 0.907, 0.167, and 42.490, respectively. G(O*CHO), or Y3, is related to X3 and X7, as shown in Equation (3B): Y3 =2354.560 × ramp (X3 − 0.177) − 2043.245 × ramp (X3 − 0.176) + 25.754 × ramp (2.033 − X7) − 1.010. Equation (3B) indicates that the average TM–N_a_ and TM–N_b_ bond length (X7) has a negative effect on G(O*CHO); a shorter bond length increases the G(O*CHO) value. While the average NBO charges of the C_a_ (X3) have a positive effect on the G(O*CHO) values—larger positive charges on the TM increase G(O*CHO). Considering the relative charges for C_a_ and the relative TM–N bond lengths in Table 1, Ni leads to higher G(O*CHO) values, while Sc gives rise to lower values, thus opposite to the case of G(H*).

Consequently, Figure 5a–c depicts the linear relationship between the QSER-model-predicted values versus the DFT-calculated values of G(H*), G(C*OOH), and G(O*CHO) for the TM–PP catalysts.

### 2.3. Newly Designed Compounds and Predicted Activities

As is commonly known, the ligands of porphyrin significantly affect its activity. For example, Savéant et al. introduced phenolic hydroxyl groups to the second coordination sphere of an Fe–porphyrin, which greatly improved the selectivity of the CO_2_RR [39]. In addition, Nichols et al. utilized second-sphere effects to design a series of iron–porphyrins to promote CO_2_ reduction selectivity [40]. In this study, a series of new TM–PP complexes were designed, introducing phenyl monoamino ligands into the porphyrin complexes; these were denoted as TM–Amino–TPPs, as shown in Figure 6. The structure was optimized using the M06L/6-31G (d, p) basis set, which is similar to that used with the training data set of the TM–PP catalysts. According to the QSER models with GA–MLR, three properties of the TM–Amino–TPPs were collected, namely, the NBO charges of TM (X1), The average NBO charges of C_a_ (X3), and the average bond length of TM–Na and TM–Nb (X7), as presented in Appendix A. These properties were then used with the models in Table 2 to predict the values of G(H*), G(C*OOH), and G(O*CHO) from the Equations (1B,2B,3B), respectively. The Gibb free energy predictions of the newly designed TM–Amino–TPPs are listed in Table 3.

Comparing the X1, X3, and X7 values of the TM–PPs and the TM–Amino–TPPs, one can note that with the introduction of amino ligands, the NBO charges of the TM and the average TM–Na and TM–Nb bond length for the porphyrins generally decrease. The TM–Amino–TPPs have less positive NBO charges on the TM and shorter TM–N bond lengths, while the amino substituent increases the average NBO charges of C_a_ for these porphyrins, except for the case of Sc–Amino–TPP. The values of the descriptors X1, X3, and X7 of the TM–PPs versus those of the TM–Amino–TPPs are plotted in Appendix A, showing that the TM NBO charges of Cr– and Mn–Amino–TPP decrease the most and the NBO charges of C of these same two porphyrins increase the most with the introduction of amino ligands. This indicates that the incorporation of amino ligands could redistribute the electrons of the porphyrin molecules; the Cr and Mn metal centers have a greater positive charge, and they can adsorb the CO_2_^−^ anion more easily.

Gibbs free energy is one of the critically important values for determining the preferable reaction pathways. Thus, in case that the G(H*) values are lower than that of G(C*OOH) and G(O*CHO), the HER reaction is preferable, while the candidate catalysts for the CO_2_ reaction pathway, their G(C*OOH) or G(O*CHO) values are lower than that of G(H*). Therefore, in the newly designed TM–Amino–TPPs based on the QSER predictions (Appendix A), the Sc–, Ti–, V–, Cr–, Co–, Cu–, and Zn–Amino–TPPs show the lower Gibbs free energies of O*CHO compared to H* and C*OOH as shown in Figure 7. Therefore, this can imply that these catalysts could favorably convert CO_2_ to HCOOH as O*CHO as an intermediate for HCOOH products. On the other hand, the Mn–, Fe–, and Ni–Amino–TPPs catalysts showed lower Gibbs free energies of C*OOH (Figure 7); thus, these new catalysts could support the reaction pathway of CO product formation.

Furthermore, considering the crucial property of the NBO charges of the metal center of TM–Amino–TPPs (Appendix A), they are significantly decreased compared to that of TM–PPs. Therefore, the charges on the TM–PPs catalysts are redistributed with the introduction of monoamino phenyl ligands, resulting in a more positive charge on the metal center, which would become a more active center. Meanwhile, it is worth noting that the QSER trendy prediction for the Fe–Amino–TPP catalyst corresponds well with the experimental work of Abdinejad et al., as the Fe–Amino–TPP catalyst exhibited a good CO_2_-to-CO conversion, with an FE of 49% [24]. Based on the three types of Gibbs free energy QSER-model predictions, the newly designed Mn–, Fe– and Ni–Amino–TPPs are selected for further reaction mechanism investigations as their predicted G(C*OOH) is lower than the G(O*CHO) and G(H*). Thus, they could benefit from catalyzing the CO_2_ to CO conversion selectivity. However, the product between the CO and HCOOH was intensively studied, and it was found that the product selectivity switched by changing the metal center [41]. Therefore, the QSER results only provide the preliminary screening for potential or candidate catalysts for further theoretical reaction mechanism investigation, which need to include the activation energy and the transition state to determine the rate-determining step of the CO_2_RR reaction.

## 3. Data and Methodology

### 3.1. Data for the Training Set

The Gibbs free energies of reaction intermediates determine the most appropriate reaction pathway and products for the CO_2_RR [7]. The lower the Gibbs’s free energy, the more likely it is that the reaction will occur. In addition, the HER competes with the CO_2_RR, which needs to be taken into consideration in predicting the reaction selectivity [42]. In this work, we used the Gibbs free energies of the CO_2_RR and HER reaction intermediates C*OOH, O*CHO, and H* as our target activities to build the QSAR models. The data on the TM–PPs were obtained from Liu et al.’s work and are partitioned into three training sets—the Gibbs free energies of the H*, C*OOH, and O*CHO intermediates—denoted as G(H*), G(C*OOH), and G(O*CHO), or Y1, Y2, and Y3, respectively, as presented in Table 4 [22].

### 3.2. Optimization Details and Properties

The frequency of all of the TM–PP and TM–Amino–TPPs complexes were calculated and subsequently optimized using DFT with the M06L exchange–correlation functional and the 6-31G (d, p) basis set in the Gaussian 09 program [43]. Based on the optimized structures, the structural and electronic properties were obtained, consisting of the natural bond orbital (NBO) charges of the TM (X1); average NBO charges of N_a_ and N_b_ (X2), C_a_ (X3), and C_b_ (X4); dipole moment (X5); polarizability (X6); average TM–Na and TM–Nb bond length (X7); and average N_a_–N_b_ distance (X8), as shown in Figure 3 and Table 1.

### 3.3. QSER Technique Used in This Work

The central idea of genetic algorithms is that the region to be searched is coded into one or multiple strings. The genetic algorithm operates with a set of such strings, termed a population [44]. In QSER studies, to build a model with high correlation and predictive ability, the choice of descriptors is notably significant [45]. Hence, we employed the GA method to effectively select the appropriate variables. As the Gibbs free energies of intermediates play an important role in predicting electrocatalytic performances, we used these values for the H*, C*OOH, and O*CHO intermediates as three dependent variables [38]. Additionally, the correlation of the three Gibbs free energies and the eight DFT properties of the TM–PP complexes have also been analyzed using the Pearson correlation coefficient. The GA–MLR technique was implemented in the Materials Studio program to derive the QSER models [46]. Initially, the training data were fully imported with the maximum equation length set at three as the limited training data; the population and the maximum number of generations were set to 3000 and 500, respectively, and the mutation probability was 0.1.

### 3.4. Statistical Terms for QSER Analysis

The performance of the regression models was measured using Fischer’s ratio (F-value); R^2^; the cross-validation R-squared, or R^2^(CV); and the residual sum of squares (RSS) [47]. The F-value reflects the ratio between the variance explained by the model and the variance due to the error in the regression; high F-values indicate that the model is statistically significant. R^2^ is a statistical measure of fit that indicates how much of the variation in a dependent variable is explained by the independent variable(s) in a regression model, as given by Equation (4). We used the leave-one-out validation technique to obtain the values of R^2^(CV), a key measure of the predictive power of a model, as calculated in Equation (5); the closer the value is to 1.0, the better the predictive power. The RSS measures the level of variance in the error term, or residuals, of a regression model, calculated according to Equation (6). Ideally, the sum of the squared residuals should be smaller than the sum of squares from the regression model’s inputs. We also calculated the root-mean-square error (RMSE) to evaluate the accuracy of our models (Equation (7)); the closer the value is to 0, the greater the prediction accuracy of the models.
(4)$$R^{2}=\frac{\sum_{i=1}^{n}{\left(y_{i}^{\prime }-\bar{y}\right)}^{2}}{\sum_{i=1}^{n}{\left(y_{i}-\bar{y}\right)}^{2}}$$
(5)$$R^{2}\left(\mathrm{CV}\right)=1-\frac{\sum_{i=1}^{n}{\left(y_{i}-y_{i}^{\prime }\right)}^{2}}{\sum_{i=1}^{n}{\left(y_{i}-\bar{y}\right)}^{2}},\;\mathrm{and}$$
(6)$$\mathrm{RSS}=\sum_{i=1}^{n}{\left(y_{i}-y_{i}^{\prime }\right)}^{2}$$
(7)$$\mathrm{RMSE}=\sqrt{\frac{\sum_{i=1}^{n}{\left(y_{i}^{\prime }-y_{i}\right)}^{2}}{n}}$$
where $y_{i}^{\prime }$ is the predicted value and $\bar{y}$ is the average of $y_{i}$.

## 4. Conclusions

In summary, the exploration of potential CO_2_RR candidate catalysts can be accelerated through molecular structural descriptors obtained from the DFT of the catalyst cluster model, and then the QSER concept. A series of transition-metal-coordinated porphyrins and their calculated CO_2_ reduction activity in terms of Gibbs free energies of H*, C*OOH, and O*CHO were used as training data for the QSER model. The genetic algorithm combined with multiple linear regression (GA–MLR) techniques was used to manipulate the QSER mathematical models of all three Gibbs free energies. Based on our QSER models, both the electronic properties (charges of the metal and porphyrin) and the structural properties (bond lengths between the metal center and the nitrogen atoms of the porphyrin) play a significant role in the three Gibbs free energies. The obtained results showed a good predictive ability both (R^2^ value) R^2^(CV), which is more than 0.8. Subsequently, QSER with GA–MLR models were then used to predict the Gibbs free energies of H*, C*OOH, and O*CHO of CO_2_RR for a series of newly designed transition-metal monoamino-substituted tetraphenylporphyrins (TM–Amino–TPPs, TM = Sc, Ti, V, Cr, Mn, Fe, Co, Ni, Cu, and Zn). The predicted three kinds of energies found that the Sc–, Ti–, V–, Cr–, Co–, Cu–, and Zn–Amino–TPPs showed lower Gibbs free energies of O*CHO compared to H* and C*OOH; therefore, they could have the potential to convert CO_2_ to HCOOH. While considering the Mn–, Fe–, and Ni–Amino–TPPs, which exhibit lower Gibbs free energies of C*OOH than others, it can be deduced that CO_2_–CO conversion selectivity occurs on these catalysts. Overall, the DFT descriptors and the QSER with GA–MLR methods can be used to estimate or screen potential catalysts for CO_2_ electroreduction.

## Figures and Tables

**Figure 1 molecules-28-03105-f001:**
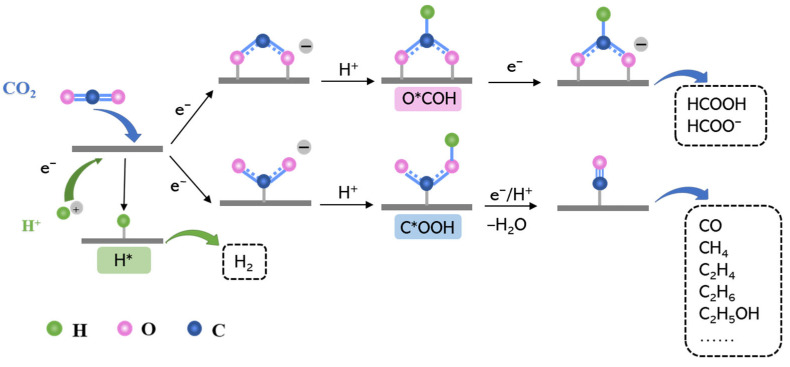
The main intermediates of C*OOH and O*COH for the CO_2_ reduction reaction (CO_2_RR) and the H* intermediate for the hydrogen evolution reaction (HER).

**Figure 2 molecules-28-03105-f002:**
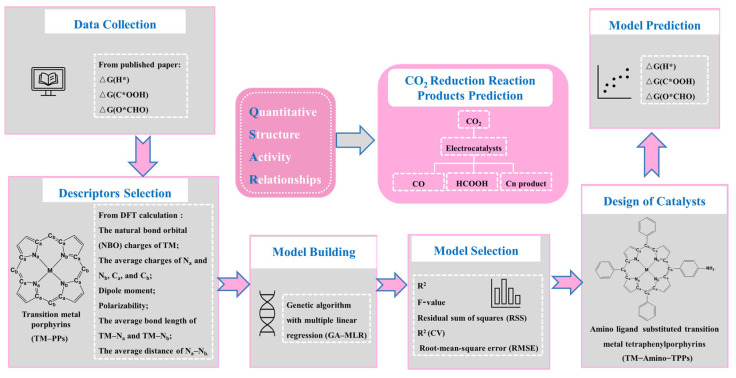
Workflow of the QSER–accelerated catalytic activity prediction of transition-metal–porphyrin complexes in this study.

**Figure 3 molecules-28-03105-f003:**
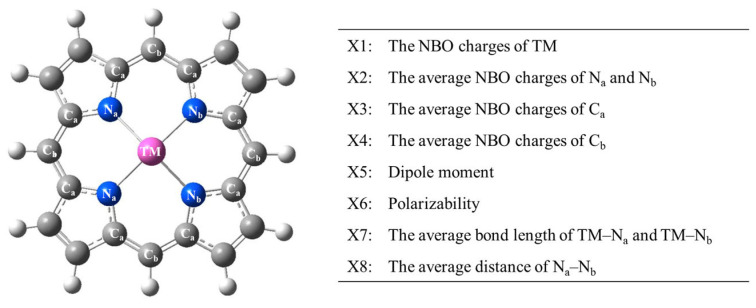
The optimized structure of the transition-metal–porphyrin complexes and the definitions of the structural and electronic properties (TM–PPs, TM = Sc, Ti, V, Cr, Mn, Fe, Co, Ni, Cu, and Zn). The white, gray, blue, and pink colors represent H, C, N, and TM atoms, respectively.

**Figure 4 molecules-28-03105-f004:**
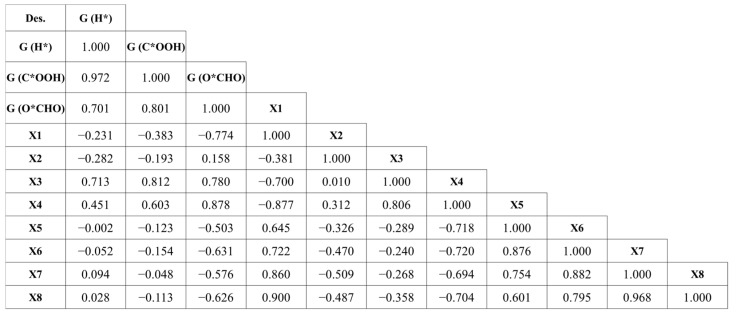
Pearson correlation coefficients between the three Gibbs free energies and the eight properties obtained from DFT of the TM–PP complexes.

**Figure 5 molecules-28-03105-f005:**
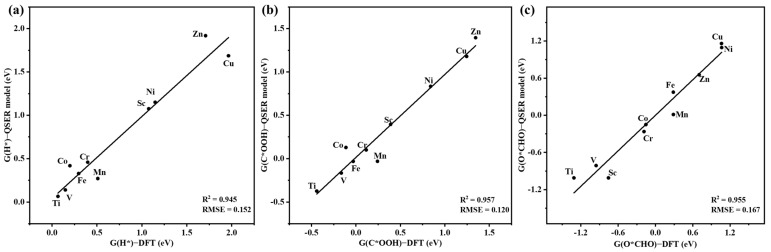
Plots of the QSER-model-predicted values versus the DFT-calculated values for (**a**) G(H*), (**b**) G(C*OOH), and (**c**) G(O*CHO) for the TM–PP catalysts using the GA–MLR method.

**Figure 6 molecules-28-03105-f006:**
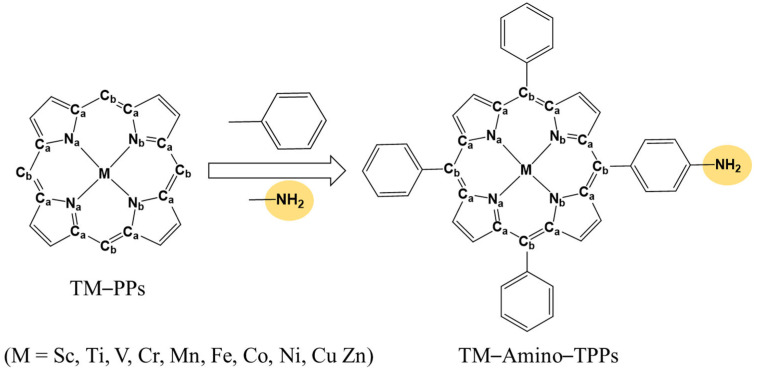
The idea of designing new TM–amino–porphyrin complexes.

**Figure 7 molecules-28-03105-f007:**
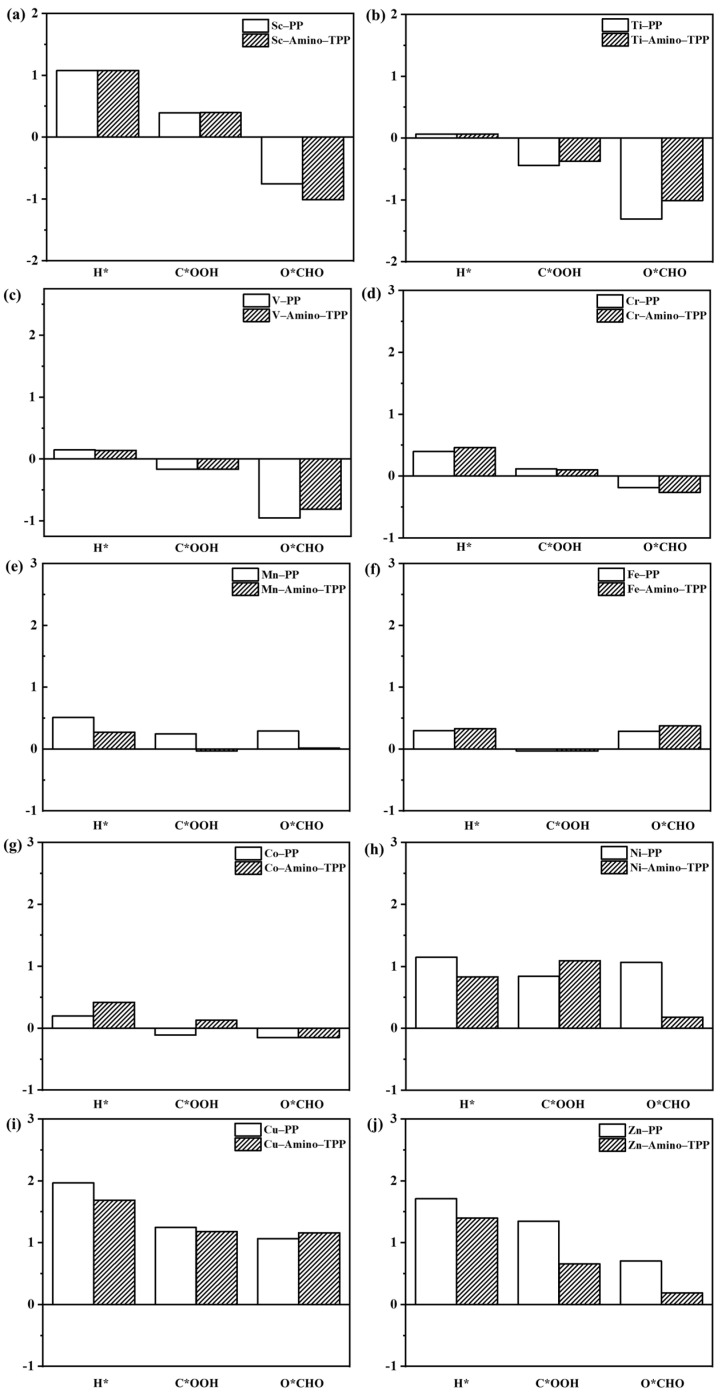
A comparison between the Gibbs free energies of the reaction intermediates for the TM–PPs and TM–Amino–TPPs, where TM is (**a**) Sc, (**b**) Ti, (**c**) V, (**d**) Cr, (**e**) Mn, (**f**) Fe, (**g**) Co, (**h**) Ni, (**i**) Cu and (**j**) Zn.

**Table 1 molecules-28-03105-t001:** Structural and electronic properties of transition-metal–porphyrins (TM–PPs).

TM–PPs Properties	Sc	Ti	V	Cr	Mn	Fe	Co	Ni	Cu	Zn
X1	1.844	1.749	1.489	1.461	1.464	1.328	1.139	0.977	1.153	1.338
X2	−0.730	−0.695	−0.662	−0.646	−0.634	−0.629	−0.601	−0.577	−0.626	−0.674
X3	0.192	0.156	0.165	0.152	0.127	0.160	0.165	0.169	0.174	0.177
X4	−0.303	−0.264	−0.249	−0.306	−0.181	−0.240	−0.238	−0.236	−0.236	−0.236
X5	1.746	0.000	0.002	0.01	0.002	0.002	0.026	0.000	0.040	0.014
X6	340.887	330.585	342.308	319.64	335.457	318.41	318.07	318.225	322.589	325.050
X7	2.107	2.038	2.024	2.008	2.002	1.983	1.966	1.949	1.998	2.034
X8	4.151	4.072	4.049	4.017	3.984	3.967	3.932	3.897	3.997	4.069

**Table 2 molecules-28-03105-t002:** Equations of the lines of best fit for predicting the values of Y1 (G(H*)), Y2 (G(C*OOH)), and Y3 (G(O*CHO)) for the TM–PPs.

Equation No.	Equation	F-Value	R^2^	R^2^(CV)	RSS	RMSE
Equation (1A)	Y1= − 5.994 × X1 − 14.18 × X3+ 35.934 × X7 − 60.834	9.089	0.820	0.603	0.754	0.275
Equation (1B)	Y1 = 383.311 × ramp (X3 − 0.183) + 29.037 × ramp (X7 − 1.953) + 33.1439 × ramp (2.069 − X7) − 3.440	34.273	0.945	0.832	0.231	0.152
Equation (2A)	Y2= − 5.850 × X1 − 17.701 × X3 + 33.585 × X7 − 56.143	12.238	0.860	0.683	0.471	0.217
Equation (2B)	Y2 = 11.025 × ramp (X1 − 1.011) + 367.618 × ramp (X3 − 0.184) + 11.838 × ramp (1.768 − X1) − 8.726	44.384	0.957	0.924	0.144	0.120
Equation (3A)	Y3= − 4.122 × X1 + 11.431 × X7 − 17.236	9.280	0.726	0.574	1.692	0.404
Equation (3B)	Y3 = 2354.560 × ramp (X3 − 0.177) − 2043.245 × ramp (X3 − 0.176) + 25.754 × ramp (2.033 − X7) − 1.010	42.490	0.955	0.907	0.278	0.167

X1, X3, and X7 represented the NBO charges of the TM, the average NBO charges of C_a_, and the average TM–N_a_ and TM–N_b_ bond lengths, respectively.

**Table 3 molecules-28-03105-t003:** The predicted values of G(H*), G(C*OOH), and G(O*CHO) from Equations (1B,2B,3B), respectively, for the newly designed transition-metal amino–porphyrin complexes (TM–Amino–TPPs, TM = Sc, Ti, V, Cr, Mn, Fe, Co, Ni, Cu, and Zn).

TM–Amino–TPPs	G(H*)/eV	G(C*OOH)/eV	G(O*CHO)/eV
Sc	1.075	0.395	−1.01
Ti	0.066	−0.373	−1.01
V	0.14	−0.166	−0.812
Cr	0.459	0.1	−0.264
Mn	0.271	−0.031	0.012
Fe	0.328	−0.032	0.373
Co	0.419	0.128	−0.15
Ni	1.15	0.834	1.094
Cu	1.687	1.179	1.159
Zn	1.918	1.395	0.654

**Table 4 molecules-28-03105-t004:** The G(H*), G(C*OOH), and G(O*CHO) values of the intermediates used as the training sets for a series of transition-metal–porphyrin complexes (TM–PPs, TM = Sc, Ti, V, Cr, Mn, Fe, Co, Ni, Cu, and Zn). Reprinted from the supporting information of Ref. [22].

TM–PPs	G(H*)/eV	G(C*OOH)/eV	G(O*CHO)/eV
Sc	1.076	0.390	−0.756
Ti	0.066	−0.441	−1.311
V	0.148	−0.165	−0.954
Cr	0.397	0.116	−0.184
Mn	0.509	0.242	0.288
Fe	0.296	−0.031	0.287
Co	0.199	−0.113	−0.153
Ni	1.148	0.840	1.065
Cu	1.964	1.245	1.062
Zn	1.709	1.346	0.702

## Data Availability

Not applicable.

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
