# Peer review of "Quantitative Structure–Electrochemistry Relationship (QSER) Studies on Metal–Amino–Porphyrins for the Rational Design of CO2 Reduction Catalysts"

_molecules, 2023, doi:10.3390/molecules28073105_

Round 1
Reviewer 1 Report
The paper focuses on the application of the QSER method to a series of porphyrin metal complexes, in order to relate their structural properties to their ability to work as catalysts in CO2 reduction. The topic is hot and a theoreticl approach able to predict electronic and structural properties of a molecule before its synthesis is surely welcome.
Therefore, I feel to recommend publication of the manuscript, after addressing some points:
1) on page 2, line 53, please add some references (such as ACS Catal. 2017, 7, 70−88, Coord. Chem. Rev. 2017, 334, 184, Front. Chem. 2019, 7, 177) at the end of the statement "Metalloporphyrin complexes have been used as catalysts for CO2 reduction on account of their unique structure and electronic properties"
2) the meaning of H*, C*OOH and O*CHO needs to be better explained
3) Figure 1 needs to be better explained
4) How is figure 4 on page 4 derived from the data? The authors should explain it
5) In table 1 there is an alignement problem in the last column (X6-Zn)
6) Startig from page 6 there is a problem with the numbering of the figures! I suggest the authors to check it carefully, also in the text
7) On page 7 the structure of the newly designed amino-porphyrin complex is not so clear. I suggest the authors to add a simple chemdraw sketch of the molecule.
8) I recommend the authors to check carefully thte main text, since there are some statements which are not so straightforward. For instance: line 213 ("As know that the HER competes with the CO2RR reaction; thus, 213 comparing"; line 215 ("Therefore, the reaction pref-215 erable route is plotted and shows in Figure 6, the values of G(C*OOH), G(O*CHO), and 216 G(H*) of all catalysts based on the obtained from QSER model predictions."); line 227 ("Another word, introduction of the mono-amino groups into the metal-porphyrins, it is not preferable for Gibb free energy of C*OOH"); line 240 ("The new designed TM-Amino-TPPs, the NBO charges of the TM significantly decrease and the charges on the metalloporphyrin molecules are redistributed")
9) the figure on page 8 should be enhanced
Author Response
Reviewer 1
- Q: on page 2, line 53, please add some references (such as ACS Catal. 2017, 7, 70−88, Coord. Chem. Rev. 2017, 334, 184, Front. Chem. 2019, 7, 177) at the end of the statement "Metalloporphyrin complexes have been used as catalysts for CO2 reduction on account of their unique structure and electronic properties"
Reply: Thank you for your kind suggestion. We added some references at end of the statement “Metalloporphyrin complexes have been used as catalysts for CO2 reduction on account of their unique structure and electronic properties [12–18]”
- Takeda, H.; Cometto, C.; Ishitani, O.; Robert, M. Electrons, Photons, Protons and Earth-Abundant Metal Complexes for Mo-lecular Catalysis of CO2 Reduction. ACS Catal. 2017, 7, 70–88.
- Di Carlo, G.; Orbelli Biroli, A.; Pizzotti, M.; Tessore, F. Efficient Sunlight Harvesting by A4 β-Pyrrolic Substituted ZnII Por-phyrins: A Mini-Review. Front. Chem. 2019, 7, 177.
- Bonin, J.; Maurin, A.; Robert, M. Molecular Catalysis of the Electrochemical and Photochemical Reduction of CO2 with Fe and Co Metal Based Complexes. Recent Advances. Coord. Chem. Rev. 2017, 334, 184–198.
- Varela, A.S.; Ju, W.; Bagger, A.; Franco, P.; Rossmeisl, J.; Strasser, P. Electrochemical Reduction of CO2 on Metal-Nitrogen-Doped Carbon Catalysts. ACS Catal. 2019, 9, 7270–7284.
- Wang, Z.; Zhou, W.; Wang, X.; Zhang, X.; Chen, H.; Hu, H.; Liu, L.; Ye, J.; Wang, D. Enhanced Photocatalytic CO2 Reduction over TiO2 Using Metalloporphyrin as the Cocatalyst. Catalysts 2020, 10, 654.
- Gotico, P.; Halime, Z.; Aukauloo, A. Recent Advances in Metalloporphyrin-Based Catalyst Design towards Carbon Dioxide Reduction: From Bio-Inspired Second Coordination Sphere Modifications to Hierarchical Architectures. Dalton Trans. 2020, 49, 2381–2396.
- Yan, T.; Guo, J.-H.; Liu, Z.-Q.; Sun, W.-Y. Metalloporphyrin Encapsulation for Enhanced Conversion of CO2 to C2H4. ACS Appl. Mater. Interfaces 2021, 13, 25937–25945.
- Q: the meaning of H*, C*OOH and O*CHO needs to be better explained and Figure 1 needs to be better explained
Reply: Thank you again for your advice. We explained the meaning of H*, C*OOH and O*CHO in our manuscript (line 67-71).
“During the CO2 reduction reaction, the first protonation step is CO2 + H+ + e- → C*OOH or O*CHO [25]. These two main intermediates (C*OOH and O*CHO) will lead to the products of CO and HCOOH, respectively [26,27]. At the meantime, hydrogen evolution reaction (HER) will also occur and it will form H* intermediates, this reaction is a competitive reaction with CO2RR [28]. The three important intermediates are shown in Figure 1.”
- Q: How is figure 4 on page 4 derived from the data? The authors should explain it
Reply: We explained how we used to derive the data of Figure 4, as following
“The correlation of the three Gibbs free energies and the eight DFT properties of the TM-PP complexes have also been analyzed by using the Pearson correlation coefficient.” (line 105-107)
- Q: In table 1 there is an alignement problem in the last column (X6-Zn)
Reply: Thank you for point out our mistakes. We have already realigned the numbers in table 1.
|
TM-PPs Properties |
Sc |
Ti |
V |
Cr |
Mn |
Fe |
Co |
Ni |
Cu |
Zn |
|
X1 |
1.844 |
1.749 |
1.489 |
1.461 |
1.464 |
1.328 |
1.139 |
0.977 |
1.153 |
1.338 |
|
X2 |
-0.730 |
-0.695 |
-0.662 |
-0.646 |
-0.634 |
-0.629 |
-0.601 |
-0.577 |
-0.626 |
-0.674 |
|
X3 |
0.192 |
0.156 |
0.165 |
0.152 |
0.127 |
0.160 |
0.165 |
0.169 |
0.174 |
0.177 |
|
X4 |
-0.303 |
-0.264 |
-0.249 |
-0.306 |
-0.181 |
-0.240 |
-0.238 |
-0.236 |
-0.236 |
-0.236 |
|
X5 |
1.746 |
0.000 |
0.002 |
0.01 |
0.002 |
0.002 |
0.026 |
0.000 |
0.040 |
0.014 |
|
X6 |
340.887 |
330.585 |
342.308 |
319.64 |
335.457 |
318.41 |
318.07 |
318.225 |
322.589 |
325.050 |
|
X7 |
2.107 |
2.038 |
2.024 |
2.008 |
2.002 |
1.983 |
1.966 |
1.949 |
1.998 |
2.034 |
|
X8 |
4.151 |
4.072 |
4.049 |
4.017 |
3.984 |
3.967 |
3.932 |
3.897 |
3.997 |
4.069 |
- Q: Startig from page 6 there is a problem with the numbering of the figures! I suggest the authors to check it carefully, also in the text
Reply: Thank you again for the mistakes. We have already carefully checked the numbering of the figures.
- Q: On page 7 the structure of the newly designed amino-porphyrin complex is not so clear. I suggest the authors to add a simple chemdraw sketch of the molecule.
Reply: Thank you for you great suggestion. We modified the Figure 5 to be 2D chemical structures of the new designed TM-Amino-porphylin as shown in line.
Figure 5. The idea of designing new TM-amino-porphyrin complexes.
- Q: I recommend the authors to check carefully the main text, since there are some statements which are not so straightforward. For instance: line 213 ("As know that the HER competes with the CO2RR reaction; thus, comparing"; line 215 ("Therefore, the reaction preferable route is plotted and shows in Figure 6, the values of G(C*OOH), G(O*CHO), and 216 G(H*) of all catalysts based on the obtained from QSER model predictions."); line 227 ("Another word, introduction of the mono-amino groups into the metal-porphyrins, it is not preferable for Gibb free energy of C*OOH").
Reply: Thank you so much. The mentioned sentences above were modified to be
“Gibbs free energy is one of the critical importance values for determining the preferable reaction pathways. Thus, in case the G(H*) values are lower than that of G(C*OOH) and G(O*CHO), the HER reaction is preferable, while the candidate catalysts for the CO2 reaction pathway, their G(C*OOH) or G(O*CHO) values are lower than that of G(H*). Therefore, newly designed TM-Amino-TPPs, based on the QSER predictions, the Sc-, Ti-, V-, Cr-, Co-, Cu-, and Zn-Amino-TPPs show the lower Gibbs free energies of O*CHO compared to H* and C*OOH. Therefore, this can imply that these catalysts could favorably convert CO2 to HCOOH as O*CHO is an intermediate for HCOOH products. On the other hand, the Mn-, Fe-, and Ni-Amino-TPPs catalysts showed lower Gibbs free energies of C*OOH; thus, these new catalysts could support the reaction pathway of CO product formation.” Line 219 - 229
- Q: line 240 ("The new designed TM-Amino-TPPs, the NBO charges of the TM significantly decrease and the charges on the metalloporphyrin molecules are redistributed")
Reply: Thank you so much. The mentioned sentences above were modified to be
“Furthermore, considering the crucial property of the NBO charges of the metal center of TM-Amino-TPPs, they are significantly decreased compared to that of TM-PPs. Therefore, the charges on the TM-PPs catalysts are redistributed with the introduction of mono-amino phenyl ligands, resulting in a more positive charge on the metal center which would become a more active center.” Line 230 - 234
- Q: the figure on page 8 should be enhanced
Reply: Thank you very much. We removed this figure.

Reviewer 2 Report
Thanks for the opportunity to review the manuscript titled, “Quantitative Structure–electrochemistry Relationship (QSER) Studies on Metal-Amino-Porphyrins for Rational Design of CO2 Reduction Catalysts” by Maitarad and co-workers. The current manuscript presented an interesting strategy for the optimization of CO2 reduction catalysts.
In my opinion, two major concerns in the current manuscript are not well addressed. The first one is about the correlation matrix / QSER model relationship. The three Gibbs free energies, G(H*), G(O*CHO), and G(O*CHO) were fitted with eight properties of 122 the TM-PP complexes. However, the Gibbs free energies of activation (DG‡) of the rate-determining step (RDS) or the Gibbs free energies of activation (DG‡) of the turnover-limiting step (TOLS) were not even discussed in the current manuscript. The conclusion obtained from the fitting based on the Gibbs free energies instead of the Gibbs free energies of activation is not convincing, and further investigations are required.
The second one is about the product selectivity and the G(O*CHO) and G(O*CHO) properties in the QSER model. Robert and co-workers (J. Am. Chem. Soc., 2015, 137, 10918. https://doi.org/10.1021/jacs.5b06535) showed that switching of the metal center could form different products for CO2 reduction ([Co(N5)](ClO4)2 yielded CO and [Fe(N5)Cl2](ClO4) yielded HCCOH). The fitting of G(O*CHO) and G(O*CHO) properties ignored this point. It is also well-known the volcano plot exists in the transition-metal catalyzed CO2RR and the HER. But Figure 6 in the current manuscript didn’t clearly present this feature.
Author Response
Reviewer 2
Thanks for the opportunity to review the manuscript titled, “Quantitative Structure–electrochemistry Relationship (QSER) Studies on Metal-Amino-Porphyrins for Rational Design of CO2 Reduction Catalysts” by Maitarad and co-workers. The current manuscript presented an interesting strategy for the optimization of CO2 reduction catalysts.
In my opinion, two major concerns in the current manuscript are not well addressed. The first one is about the correlation matrix / QSER model relationship. The three Gibbs free energies, G(H*), G(O*CHO), and G(O*CHO) were fitted with eight properties of 122 the TM-PP complexes. However, the Gibbs free energies of activation (DG‡) of the rate-determining step (RDS) or the Gibbs free energies of activation (DG‡) of the turnover-limiting step (TOLS) were not even discussed in the current manuscript. The conclusion obtained from the fitting based on the Gibbs free energies instead of the Gibbs free energies of activation is not convincing, and further investigations are required.
Reply: Thank you so much reviewer for your kind reviewing and suggesting
Your suggestion is so valuable, as in the first manuscript, we implied the results over the scope of our obtained QSER Gibbs free energy prediction. Therefore, the revised manuscript, part of 2.4 has been removed from the revised manuscript. The discussion for a series of newly designed transition-metal monoamino-substituted tetraphenylporphyrins (TM-Amino-TPPs, TM = Sc, Ti, V, Cr, Mn, Fe, Co, Ni, Cu, and Zn) has been combined into the 2.3 Newly designed compounds and predicted activities. (Line 219 – 246)
Gibbs free energy is one of the critical importance values for determining the preferable reaction pathways. Thus, in case the G(H*) values are lower than that of G(C*OOH) and G(O*CHO), the HER reaction is preferable, while the candidate catalysts for the CO2 reaction pathway, their G(C*OOH) or G(O*CHO) values are lower than that of G(H*). Therefore, newly designed TM-Amino-TPPs, based on the QSER predictions, the Sc-, Ti-, V-, Cr-, Co-, Cu-, and Zn-Amino-TPPs show the lower Gibbs free energies of O*CHO compared to H* and C*OOH. Therefore, this can imply that these catalysts could favorably convert CO2 to HCOOH as O*CHO is an intermediate for HCOOH products. On the other hand, the Mn-, Fe-, and Ni-Amino-TPPs catalysts showed lower Gibbs free energies of C*OOH; thus, these new catalysts could support the reaction pathway of CO product formation.
Furthermore, considering the crucial property of the NBO charges of the metal center of TM-Amino-TPPs (Figure S2), they are significantly decreased compared to that of TM-PPs. Therefore, the charges on the TM-PPs catalysts are redistributed with the introduction of mono-amino phenyl ligands, resulting in a more positive charge on the metal center which would become a more active center. Meanwhile, it is worth noting that the QSER trendy prediction on the Fe-Amino-TPP catalyst corresponds well with the experimental work of Abdinejad et al., as the Fe-Amino-TPP catalyst exhibited a good CO2-to-CO conversion, with a FE of 49% [26]. Based on the three types of Gibbs free energy QSER model predictions, the newly de-signed Mn-, Fe- and Ni-Amino-TPPs are selected for further reaction mechanism investigations as their predicted G(C*OOH) is lower than the G(O*CHO) and G(H*). Thus, they could benefit for catalyzing the CO2 to CO conversion selectivity. However, the product between the CO and HCOOH was intensively studied, and it found that the product selectivity switched by changing the metal center. Therefore, the QSER results only provide the preliminary screening for potential or candidate catalysts for further theoretical reaction mechanism investigation, which need to include the activation energy and the transition state to determine the rate-determining step of the CO2 RR reaction
The previous Figure 6 is removed from this revised manuscript.
The second one is about the product selectivity and the G(O*CHO) and G(O*CHO) properties in the QSER model. Robert and co-workers (J. Am. Chem. Soc., 2015, 137, 10918. https://doi.org/10.1021/jacs.5b06535) showed that switching of the metal center could form different products for CO2 reduction ([Co(N5)](ClO4)2 yielded CO and [Fe(N5)Cl2](ClO4) yielded HCCOH). The fitting of G(O*CHO) and G(O*CHO) properties ignored this point. It is also well-known the volcano plot exists in the transition-metal catalyzed CO2RR and the HER. But Figure 6 in the current manuscript didn’t clearly present this feature.
Reply: Thank you for the literature suggestions; the reference is cited in Line 242. Based on the QSER predictions, three catalysts of Mn-, Fe- and Ni-Amino-TPPs are selected for further reaction mechanism investigations, including the activation energy and the transition state, to determine the rate-determining step of the CO2 RR reaction. Furthermore, product selectivity switching will also consider. As the part of the DFT reaction mechanism study will be time-consuming and cluster resources limitation, therefore, in this manuscript, we only can present the part of QSER first. The DFT reaction pathway will be further performed to confirm our QSER prediction.

Round 2
Reviewer 2 Report
The revised manuscript addresses the reviewer's concerns and is acceptable for publication.